# Between Resilience and Agency: A Systematic Review of Protective Factors and Positive Experiences of LGBTQ+ Students

**DOI:** 10.3390/healthcare11142098

**Published:** 2023-07-23

**Authors:** Telmo Fernandes, Beatriz Alves, Jorge Gato

**Affiliations:** 1Centre for Psychology, University of Porto, 4099-002 Porto, Portugal; jorgegato@fpce.up.pt; 2Faculty of Psychology and Education Sciences, University of Porto, 4099-002 Porto, Portugal

**Keywords:** minority stress, risk, resilience, protective factors, LGBTQ students, systematic review

## Abstract

A negative school climate resulting from homophobic and transphobic bias and discrimination is associated with poor well-being and mental health among LGBTQ+ youth. However, protective factors and mechanisms may buffer against the impact of stigmatization. Drawing on the socio-ecological model, minority stress theory, and positive youth development and agency perspectives, we carried out a systematic review of research focusing on factors that can promote the well-being of LGBTQ+ students in educational settings, outlining the primary outcomes from studies published between 2012 and 2022. The PRISMA protocol was used for this review, and 64 articles were scrutinized. The results of the thematic analysis revealed that both external factors (school-inclusive policies and extracurricular activities; social support from school, family, and the community; and school connectedness) and internal factors (psychosocial characteristics and personal agency) promote positive school experiences, such as the exploration of sexual and gender identities in a safe environment. The present findings highlight the need for inclusive school policies and strategies and individual-level interventions that target the well-being and positive mental health outcomes of sexual and gender minority students.

## 1. Introduction

Homophobic and transphobic bullying in school compromise the safety, sense of belonging, school achievement, and school attendance of youths who identify as lesbian, gay, bisexual, trans, queer, or as other sexual and gender minority identity (LGBTQ+), jeopardizing their well-being and mental health [1,2,3]. Transgender and gender non-conforming youth are affected in specific ways, mainly relating to their gender affirmation in a cisnormative educational environment [4]. Thus, according to the United Nations Educational, Scientific and Cultural Organization [5], education stakeholders should prioritize the challenges faced by LGBTQ+ youths in schools. In this respect, an increasing number of laws, policies, and school programs designed to foster the well-being and social acceptance of LGBTQ+ students have recently been implemented in many countries with proven efficacy [6,7]. Besides these contextual measures, sexual and gender minority youths may develop personal resources that function as protective mechanisms by compensating for, protecting them from, or challenging discrimination [8]. Such individual-level protective mechanisms include social–emotional competencies or the exploration of positive sexual identities and self-esteem [9].

Our goal in the present work was to summarize the main findings from the international scientific literature focusing on protective factors and positive experiences of LGBTQ+ students in the school context. 

Protective factors and mechanisms that act as buffers against the impact of stigma and promote the well-being of LGBTQ+ youths in schools have, in fact, been considered by multiple theoretical frameworks, namely the social-ecological model [10], the minority stress model [11,12], and positive youth development [13,14] and agency [15,16,17] approaches.

### Goals of the Present Systematic Review

An initial review of the studies concerning the protective factors and positive experiences that act as buffers against the impact of stigma and promote the well-being and mental health of LGBTQ+ youths in schools revealed that different quantitative and qualitative methodologies were used. Therefore, a systematic review was deemed the most appropriate strategy to summarize the main findings from the international scientific literature focusing on this subject. 

In the last decade, evidence of higher levels of acceptance of LGBTQ+ youths can be found, and more research is being carried out. Therefore, wider and more diverse samples and topics are being studied [6,18]. Simultaneously, state-level backlash and negative attitudes towards LGBTQ+ youths have risen in some regions [19,20,21,21], and the need to identify effective strategies to promote protective factors and safe and positive experiences in educational settings has become more urgent. Therefore, it was considered appropriate to focus our research on articles published between 2012 and 2022.

The current study aimed to respond to the following research question: what is the evidence of protective factors and positive experiences of LGBTQ+ youths in the school context in the published research? We have identified three main objectives: (1) to identify current protective factors for LGBTQ+ students in the school context; (2) to collect positive experiences of LGBTQ+ students in the school context; and (3) to explore evidence of agency in LGBTQ+ students in the school context.

## 2. Materials and Methods

The systematic review approach provided a strategy to summarize the main findings from the research produced in the delimited time frame of the last decade in different geographical and cultural contexts [22,23], following a structured and controlled process [24]. We adopted the Preferred Reporting Items for Systematic Reviews and Metanalysis (PRISMA) guidelines with a PRISMA checklist [22,25]. Pre-defined protocols such as PRISMA provide transparency to the process of systematic reviews, reducing the risk of biases and enabling the reproducibility of results [25]. This review was registered through PROSPERO (registration number CRD42023438506).

To examine and summarize the results of the systematic review, the research team opted for a thematic analysis. 

### 2.1. Search Strategy

The first stage of article selection was conducted by two independent researchers, the main author and a junior researcher. Both executed systematic research on the following eight online databases indexed to Web of Science and included on EbscoHost: Academic Search Ultimate, APA PsycInfo; Education Source; Sociology Source Ultimate; ERIC; Psychology and Behavioral Sciences Collection; APA PsycArticles; Teacher Reference Center; and Fonte Académica. These databases include publications and research focused mainly on psychology, sociology and educational sciences.

Both researchers used the following set of words in their research, selecting the field search “abstract”, in peer-reviewed academic publications only: “lgbtq OR lesbian OR gay OR homosexual OR bisexual OR transgender OR homosexual OR queer OR sexual minority OR gender minority” AND “positive identity OR positive OR coping OR coping strategies OR coping mechanisms OR cope OR protective factors OR resilience” AND “school OR school system OR education OR education system OR learning OR students OR K12 OR pupils” NOT “systematic review OR meta-analysis OR literature review OR review of literature”. 

### 2.2. Inclusion Criteria

The following inclusion criteria were established for the selection of articles: (i) articles should discuss results from empirical studies; (ii) articles should have been peer-reviewed; (iii) research should focus on at least one protective factor or positive experience of LGBTQ+ students; (iv) the sample should include self-identified LGBTQ+ students aged 14 to 19 years old; and (v) the article should have been published between January 2012 and March 2022. 

### 2.3. Study Screening

After conducting the search, 489 articles were found through EBSCOhost. The platform automatically eliminated duplicated results. Five additional articles were added, resulting from references included in some of the screened articles. A checklist template was created to assess the presence of the following features: title, country of origin, and year of the study; being an empirical study; the assessment and type of protective factor or positive experience of LGBTQ+ students; the inclusion in the sample of LGBTQ+ students from secondary schools; the study methodology; and finally, the main outcome related with this systematic review’s purpose. 

The main author and the junior researcher screened all the articles’ abstracts to decide whether they should be included. Some articles had to be read more thoroughly to decide their eligibility. The results of the two independent screenings were compared and discussed until a consensus was reached. A total of 64 studies were selected for the review (Figure 1). The country of origin of most of the studies was the United States of America, with 44 articles. Canada and the United Kingdom were the second most frequent countries, with four studies each, followed by Australia, Iceland, and South Africa (with two studies each, and one of them shared by Iceland and South Africa), and finally, China, Israel, the Netherlands, Taiwan, Scotland, and Portugal, with one study each. 

### 2.4. Data Collection

Mendeley Reference Manager software was used to organize the records and individual references. A template was created to verify in each study: (1) title, year, and country of publication; (2) protective factors and positive experiences for LGBTQ+ students; (3) the inclusion in the sample of 14- to 19-year-old LGBTQ+ students; and (4) the type of study/methodology. Then, the main author and the junior researcher read the articles’ abstracts and extracted the relevant information to fill the template. The results were compared until a consensus was achieved as to which articles should be excluded and included in the review. Full texts of the included articles were extracted from different platforms and databases.

An initial review of the studies revealed that different quantitative and qualitative methodologies were used. The main findings from each study were extracted and organized to identify the main outcomes. Thematic analysis was considered to be a more adequate approach to interpreting the results. This analytical approach can provide useful information generated through a previous theoretical frame, but it can also be flexible enough to articulate a more pre-determined frame with more reflexive and subjective insights, thus allowing previously unforeseen themes to emerge from the data [26]. After an initial reading to become familiar with the articles, preliminary codes were created and then used to identify common themes. The articles were then analyzed according to these themes, which were then reviewed, discussed by the two authors, and labeled. In almost all articles, more than one theme was identified.

Research that did not focus on protective factors or positive experiences was excluded. Additionally, articles that examined the role of protective factors and positive experiences of LGBTQ+ youths but not specifically in a school or educational context and research that used samples that did not include LGBTQ+ students between 14 to 19 years old were not considered. A total of 430 articles were excluded. 

## 3. Results

In this section, we demonstrate the different steps of the review, starting with the coding process of selected studies, followed by an overview of the main methodologies used and a brief characterization of the samples. Finally, we include a section with the results and a discussion of the themes and connected subthemes. 

### 3.1. Coding of Studies

After the exclusion of articles following the pre-determined criteria, 64 studies were analyzed. A second template was created to extract different types of protective factors, positive experiences, and other outcomes, as can be seen in Appendix A (see Appendix A).

### 3.2. Studies’ Methodologies

Most of the studies (n = 43) used a quantitative analysis approach to survey primary data collection or secondary analysis of previously collected data from large-scale surveys, such as regional or national studies (Figure 2). Most studies used cross-sectional data (n = 59), and only five used longitudinal data, with intervals spanning from 1 to 14 school years. In turn, qualitative studies (n = 21) included interviews, focus groups (in person or by phone), ethnographic research, classroom recordings, document collection, observations, and data analysis, such as narrative, thematic, or interpretative phenomenological analysis.

### 3.3. Samples

The sample sizes varied in number from 1 (1 key informant) to 21,953 self-identified LGBTQ+ students. A total of 23 articles included only sexual minority students (lesbian, gay, bisexual, queer, or questioning). Seventeen articles focused only on gender minority students (trans, queer, or gender non-conforming or gender diverse), and six articles included participants from Gay-Straight Alliances (GSAs) or other similar formats of school extracurricular projects dealing with gender and sexual diversity (which can include allies, other than sexual and gender minority youths), and the remaining sixteen articles included both sexual and gender minority students in the sample (Figure 3). 

Sixteen articles did not mention ethnic diversity in the sample, while the others did, but in most cases, the majority of the participants self-identified as White/Caucasian. Forty-five articles featured comparative data between LGBTQ+ and cisgender and heterosexual participants. Twenty articles used the secondary analysis of previously collected data through national or regional surveys on subjects like adolescent well-being, education, or sexual health. Only nine articles mentioned that parental consent was needed, whereas two stated that only adult school staff approval was required. In the latter two cases, a waiver for parental or adult consent was approved, considering the potential risk of exposure. Data collection occurred mainly in urban areas, but twelve studies explicitly mentioned the inclusion of participants from smaller cities, suburban, or rural areas.

### 3.4. Synthesis of Themes

Throughout the reviewed articles, some common themes that are linked with protective factors and positive experiences of LGBTQ+ students were identified. In the group of external protective factors, three main types of themes were identified: the first type of themes is associated with schools, and it includes “extracurricular activities” and “inclusive school policies”; the second type concerns social support-related factors, which include “family support”, “school support” and “community support”; third, the theme of “school connectedness”. In the group of internal protective factors, “personal assets” and “personal agency” emerged as the two main common themes, with “queering sexual and gender norms” and “outness and visibility” as subthemes of the latter. Figure 4 features a thematic tree with the main themes and their connections to the theoretical analysis. 

### 3.5. Extracurricular Activities

Within the group of studies that analyzed the role of the school climate, extracurricular activities that occur within and with the support of schools have proven to be effective in the prevention of negative outcomes and in promoting personal and collective skills among LGBTQ+ students. These practices include GSAs or some other form of LGBTQ+-inclusive school clubs [27,28,29,30,31,32,33,34,35,36,37,38], theatre-based activities, or some form of storytelling or narrative approach [39,40,41], mindfulness or meditation activities [42,43], and sports practice and arts [43]. Other identified extracurricular activities comprehend alternative pedagogic approaches, such as peer education [41], narrative approach [39], or critical dialogue and writing exercises [44].

### 3.6. Inclusive School Policies

Another theme identified in the reviewed articles was designated as inclusive school policies, which comprise inclusive school curricula [45], inclusive sex education activities [46], inclusive resources and spaces [47], or a set of combined and tailored LGBTQ+-inclusive policies [48,49,50,51,52]. 

### 3.7. Social Support: Family, School, and the Community

Going beyond the school context, being part of a wider context that provides social support was associated with less bullying and more room for visibility, positive well-being, and mental health [28,53]. In some cases, perceived support is derived from the school itself [54,55], and in others, a particular emphasis is placed on aspects such as supportive counseling services [56,57], support from peers [47,58], and support from teachers [59]. Social support can also result from the acceptance of family members and the possibility of being open in that context [28,54,55,60,61,62,63,64,65]. 

Support can also result from community-inclusive dimensions, such as the availability of supportive organizations and other resources, a more progressive socio-political climate, and other features, such as being surrounded by a positive urban culture [32,66]. To a lesser extent, access to positive role models was identified as a contributor to positive experiences, either through positive representation in the media [28,57] or the presence of other LGBTQ+ people in the school context [32,57].

### 3.8. School Connectedness

The theme of school connectedness was strongly associated with positive experiences for LGBTQ+ students and a common feature of positive school climates, with some evidence regarding supportive adults and an environment in which being out is safe [27,31,38,47,54,60,67,68,69,70,71,72,73], as well as room to build a sense of community among minority students and their allies [74]. 

### 3.9. Psychosocial Characteristics 

Another group of articles analyzed the role of students’ psychosocial characteristics as features that provide protection in the school context. That involves leadership skills, via assumed roles in GSAs or other youth groups [33,75], critical consciousness and self-confidence [41], self-acceptance, and self-perception as unique [28] or appreciation for one’s own differences [31] but also help-seeking behaviors [58]; performing self-advocacy [59]; being able to engage in emotional coping, such as downplaying and ignoring stigma or using humor as a response [76,77,78]; self-compassion [79]; and being able to express feelings in an assertive way [80]. Values such as personal faith in inclusive religions [28], spirituality [58], hope for the future, inspiration [31,45,81], or trust [82] were also identified as protective characteristics for this population. 

### 3.10. Personal Agency 

Another identified theme was personal agency [35,36,37,77,83] or the capacity to pro-actively own one’s identity and somehow use it to contribute to social and cultural change towards acceptance of sexual and gender diversity in the school context [41,74]. In some research, agency is expressed through acts of advocacy, including standing up for one’s rights [59] and activism, which are associated with well-being and positive experiences for these students [28,36,37,49,76,83]. 

### 3.11. Queering Sexual and Gender Norms

A subtheme associated with personal agency and positive experiences of minority youths in the school context is the variety of ways by which they express themselves through queering sexual and gender norms, either by displaying identity flexibility [28,84], resisting sexual labels [49], criticizing heteronormativity or cisnormativity [74,76], or not fulfilling gender role expectations [77,84]. 

### 3.12. Outness and Visibility

Linked with the previous theme, outness and visibility are present in LGBTQ+ students’ reports on several reviewed articles, closely associated with a sense of authenticity and pride [28,52,85,86] and positive coming out experiences [32,62,87].

## 4. Discussion

Throughout the present systematic review, we aimed to explore updated knowledge on the experiences of LGBTQ+ youths in school contexts and how both individual-level and environmental components can foster a positive climate and personal well-being. Two types of protective factors were identified: external and internal. 

According to the socio-ecological perspective [10], individual behaviors are not only the result of personal traits and assets but a result of environmental factors that articulate with individuals’ experiences. Therefore, the assessment of dimensions such as the inclusion of positive and proactive measures to tackle homophobic and transphobic bias in schools, as well as other ways of improving the safety and well-being of sexual and gender minority students, are as equally important as the research that focuses on personal assets and individual responses to bullying and discrimination [88,89]. The contribution of inclusive policies and strategies that tackle homophobia and transphobia in a consistent and effective way in the school context has been highlighted in the literature [90,91,92,93]. In particular, strategies that put an emphasis on youth capability and activity, peer support, and school adjustment have proven to have a positive and effective role in youths’ well-being [94,95]. It is therefore important to investigate the main strategies developed by different stakeholders, such as state-level institutions and NGOs, as well as school-level policies and their main outcomes in terms of school climate and LGBTQ+ youths’ well-being. 

The minority stress theory postulates that beyond general stressors, LGBTQ+ people face specific stress experiences on a continuum from distal to proximal factors [11,12]. Prejudice events such as harassment, institutional discrimination, violence, or rejection constitute distal stressors. The expectation of discrimination as a result of stigma, concealing one’s identity, and the internalization of prejudice are defined as proximal stressors [11,96]. Stigmatizing stress experiences have an impact on the well-being and mental health of sexual and gender minority youths [97,98], and this is particularly worrisome considering that the average age of coming out is now lower than in previous generations [6,92,99]. 

The minority stress model also postulates the existence of protective or buffering factors and can thus be considered a resilience model [12]. Within this framework, resilience is perceived as the capacity to cope with discrimination and a negative environment, successfully adapt to the situation, and even withdraw positive outcomes and thrive despite adversity. Besides coping, two important protective factors are social support and “community connectedness.” Studies have indeed shown that protective mechanisms for LGBTQ+ youths in schools include factors such as social support from teachers [100], as well as the presence of inclusive policies [7,93], namely in sex education activities [101,102] or policies directly focusing on tackling transphobic bullying [91]. In turn, positive relationships within the family and support from peers can function as effective protective mechanisms, promoting self-esteem and buffering the impact of homophobic victimization and internalized homophobia, as well as suicidal thoughts [2,8,103,104].

In accordance with Brofenbrenner’s ecological systems theory [10], two subtypes of external protective factors associated with the school microsystem were identified: inclusive school policies and extracurricular activities. The second sub-group includes external protective factors that are linked with social support, both from school elements (such as teachers and other school staff, supportive counselors, and colleagues), from family, and within the community (the school environment but also its surrounding context, namely the presence of inclusive resources and access to positive role models, as well as a supporting local culture, including state-level policies). School connectedness is another theme that translates into an overall positive school climate and is in itself a protective dimension. These are protective elements that concur with the minority stress model [11,12] vision of factors that promote resilience in the presence of risk factors such as social and cultural bias and discrimination. 

A contemplation of protective factors would be incomplete without the consideration of youths’ self-regulation and self-determination capacities [15]. In this sense, the positive youth development approach [13,14] and the concept of agency [16,17] are two useful frameworks. While it acknowledges the many negative aspects and challenges faced by vulnerable youths, the positive youth development approach “resists conceiving of the developmental process mainly as an effort to overcome deficits and risk”, focusing instead on the “vision of a fully able child eager to explore the world, gain competence, and acquire the capacity to contribute importantly to the world” [13] (p. 16). This perspective posits that human development is not merely a response to the environment but results from the potential of internal assets, including values and skills, that contribute to promoting a positive identity [13]. For instance, Riggle et al. [105] identified aspects of positive self-identification in transgender individuals’ narratives that go beyond self-acceptance, usually seen as the final stage of LGBTQ+ identity development [106,107]. These aspects include congruency of self, enhanced interpersonal relationships, personal growth and resiliency, increased empathy, a unique perspective on both sexes, being beyond the sex binary, increased activism, and connection to the LGBTQ+ community [105]. In addition, the benefits of positive youth development programs and interventions on students’ well-being have also been documented [108].

Agency designates individuals’ capacity to look outward and interfere with their socio-cultural environment and thus can be used to understand links between positive identity and behaviors. According to Bandura [17,109], personal agency depends on the belief in personal efficacy, in self-enhancing or self-debilitating ways, but it can also be inspired by collective aspirations and thus be framed under a collective agency perspective. For example, research has revealed that LGBTQ+ inclusive school clubs (such as GSAs) are spaces where participation, advocacy, and activism are nurtured and contribute to youth’s capacity and agency [83,95].

A second group of themes is associated with internal protective factors. Research that focused on individual coping mechanisms was clustered within the theme of psychosocial characteristics. According to the minority stress model, these assets are fundamental individual protective factors that help to cope with stigma. The articles that focused on the aspects and experiences of personal agency can be linked to previous research and theoretical framing on the concept of agency [16,17] and personal youth development [13,14]. Notably, human agency cannot be analyzed without understanding the role of the environment, and thus, internal assets should be conceived as a set of skills and behaviors that individuals use in the contexts where they interact, namely schools [17].

Both these models highlight positive dimensions of adolescents’ experiences and their role in personal growth and enhancing capacities to overcome life’s current and future challenges. For sexual and gender minority youths, being out and visible, questioning gender norms, and expressing their uniqueness is an internal protection factors since research has shown the link between being out and positive adjustment [110]. These experiences and positive displays of identity are important counter-narratives that oppose the traditional research focus on negative experiences. They also seem to concur with the fact that shifting social and cultural perceptions and attitudes concerning gender and sexuality and more positive social climates towards LGBTQ+ individuals [19] are giving way, given determined protective factors, to the exploration and positive expressions of previously stigmatized identities that need to be considered [105,111,112].

## 5. Conclusions

Conclusions drawn from the present systematic review should take into consideration some of the limitations of the reviewed studies. First, most of the studies used samples from urban backgrounds, where participants were recruited through different LGBTQ+ media and resources. LGBTQ+ students in smaller cities and rural areas have far less access to these channels, and therefore, reach-out efforts should be made in future research to include the participation of less represented minorities. Second, the majority of the studies were conducted in the USA and other English-speaking regions, as well as in countries where, to some level, sexual and gender minorities’ rights are recognized in the law. In contrast, there is a deficit of studies from regions where these rights are not guaranteed or where there is strong evidence of open persecution of LGBTQ+ people. Therefore, more studies should be conducted in other regions, with the scope of highlighting different realities and the role of cultural backgrounds and sets of values and laws in sexual and gender minority students’ well-being. Third, although many articles state that the data collection took ethnic diversity into consideration, the samples feature predominantly self-identified White individuals. 

Due to prejudice and discrimination, sexual and gender minorities are a hard-to-reach population, and thus, various sampling strategies have been used in research [113]. Historically, in the cities, sexual and gender minorities find safe spaces where they can openly explore their identities and avoid persecution. This is due to the anonymity that these contexts provide but also because it is in the cities that specific resources, such as venues, community centers, services, and events, aimed at LGBTQ+ people, can be found. This is especially true in countries and regions with more inclusive laws protecting against discrimination. LGBTQ+ youths are, therefore, easier to reach through these resources in urban contexts. Conversely, data collection among minorities is much more challenging outside of big cities and in countries with no law protection or even institutionalized persecution of minorities. These factors combined create a bias in the knowledge of sexual and gender minority students’ lives that is important to take into consideration and should motivate additional research efforts and sampling strategies.

Nevertheless, the range of samples and the diversity of methodologies included in the reviewed articles provide a rich and diverse contribution to the knowledge on protective factors and positive experiences of sexual and gender minority students in the last decade and highlight good practices that foster positive school climates and provide a protective environment where these youths can cope with discrimination in positive and vibrant new ways that need to be taken in consideration in the design of future research and policies. In particular, research efforts should be made in order to learn more about the role of individual-level factors, such as individual psychosocial resources, but also focusing on the personal agency and positive experiences of LGBTQ+ youths currently occurring in school context. In addition, policymakers should take ensure the enforcement of inclusive laws and policies that aim at fostering positive school climates, such as the investment in awareness raising and training for teachers and other school staff, but also resources that allow the planning of extracurricular activities tailored for the specific needs of LGBTQ+ students on a local level.

## Figures and Tables

**Figure 1 healthcare-11-02098-f001:**
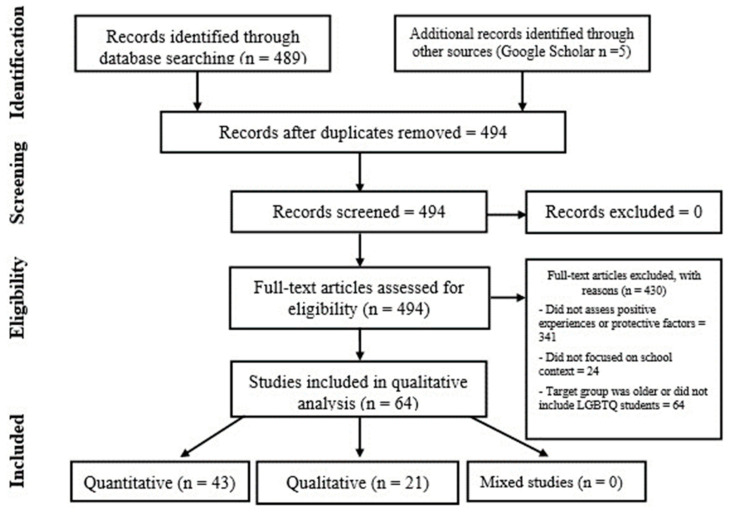
PRISMA flow chart. Exclusion criteria: (1) did not assess positive experiences or protective factors; (2) did not focus on the school context; and (3) sample included only older students or did not include LGBTQ+ students.

**Figure 2 healthcare-11-02098-f002:**
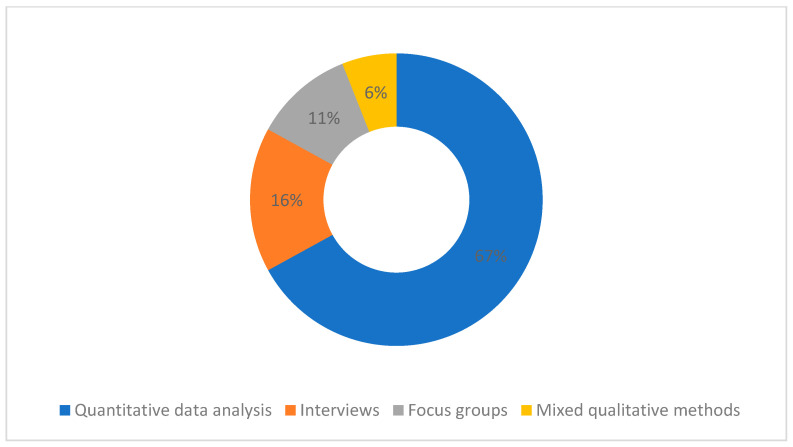
Percentage of studies’ according to methodologies.

**Figure 3 healthcare-11-02098-f003:**
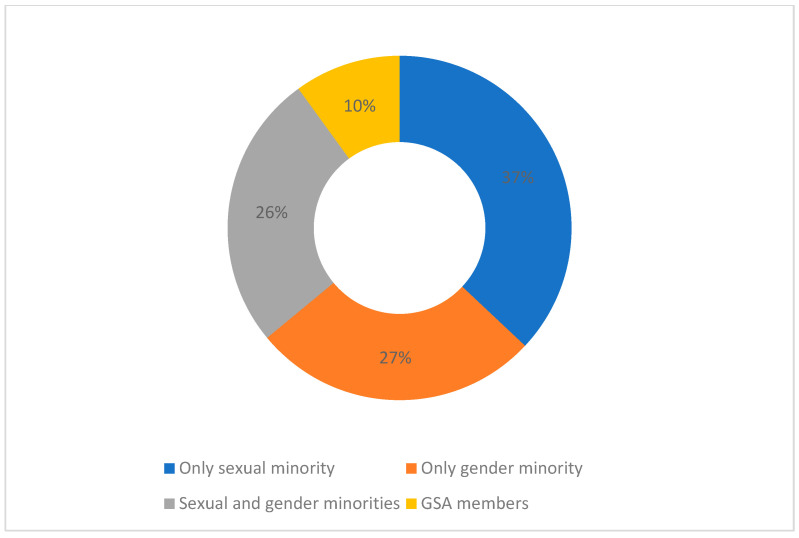
Types of studies according to sexual orientation, gender identity, and GSA membership.

**Figure 4 healthcare-11-02098-f004:**
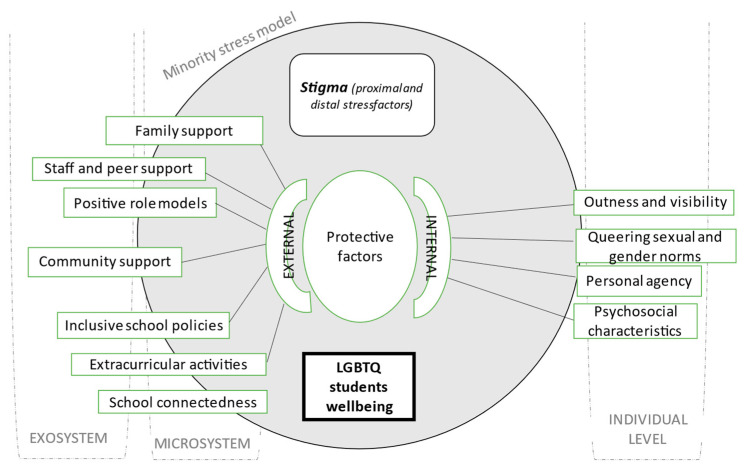
Thematic tree considering Meyer’s minority stress model [11,12] and the social-ecological framework [10].

## Data Availability

Not applicable.

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
