# Peer review of "Between Resilience and Agency: A Systematic Review of Protective Factors and Positive Experiences of LGBTQ+ Students"

_healthcare, 2023, doi:10.3390/healthcare11142098_

Round 1
Reviewer 1 Report
MAJOR ISSUES
1. Was the systematic review protocol registered? Please provide proof of registration in PROSPERO before the systematic review was conducted.
2. What is the PICO of the systematic review?
3. The objectives of the systematic review need to be enumerated. At present the authors have made general statements but these are not as concise as necessary in a systematic review.
4. Could the authors explain the absolute necessity for the presented theoretical frameworks? What purpose do these serve? In a systematic review, the authors are seeking to synthesize data found in the extant literature, not present a theoretical framework. I suggest the authors relook this section and decide if it should be retained on deleted. Additionally, if the authors have to use the theoretical frameworks, what purpose do they serve in the analysis of the data?
5. What can be learned from the analysis of the authors' findings? At present, the academic value of the conclusions remain questionable.
6. This manuscript reads more like a literature review instead of a systematic review; granted, the authors used the PRISMA guidelines but there is technically a mismatch between the intention of the authors and the outcome produced.
The use of the English Language is good, watch out for minor items, i.e. "Conclusions to be drawn", and not "Conclusions to be withdrawn".
Reviewer 2 Report
I thoroughly enjoyed reviewing this manuscript and only have some minor requests for revision. The article is well written and extremely clear. The topic is interesting, and I think this review can help to better understand how to improve the well-being of LGBTQ+ students. The whole process of finding and selecting articles is transparent and easy to follow. The discussion section adequately summarizes the results, making it easy to identify what the main protective factors are according to different areas (internal and external).
I just have some minor suggestions in order to improve the manuscript:
1. Results: although the results section is clear, a graphical representation of them would help to follow the discussion better, especially for the following subsections: studies’ methodologies and Samples. The graphical representation should show the percentage of articles for each method and for each sample.
2. Conclusions: although I agree with the limitations you have highlighted, it should be further emphasized why these studies focused only on subjects from urban centers and from countries where their rights are recognized from the low. Which are the difficulties in conducting a research in those circumstances.
Reviewer 3 Report
The manuscript entitled “Between Resilience and Agency: A Systematic Review of Protective Factors and Positive Experiences of LGBTQ+ Students” proposes a systematic review of international research focusing on factors that can promote the well-being of LGBTQ+ students in educational settings.
Through a clear and careful discussion, it presents the results of analysis drawing on the socio-ecological model, minority stress theory, and positive youth development and agency perspectives. The article makes an appreciable contribution to the study of the problem. Together with the pertinence of the theoretical perspectives used, it also notes the attention given to and clarity in the presentation of methodological choices.
While not compromising the relevance of proposals for future studies on the subject, the decision to focus attention on the articles published January 2012 to March 2022 remains unclear and substantiated. Therefore, the current proposal could benefit from a revision, with the aim of giving a more complete account of the inclusion criteria established for the selection of articles, and the indication of any limits placed by the choices made.
Round 2
Reviewer 1 Report
None